# Ocular Implications in Patients with Sleep Apnea

Nicoleta Anton [1], Roxana Elena Ciuntu [1,*], Dorin Chiseliţă [2], Ciprian Danielescu [1], Anisia Iuliana Alexa [1], Alina Cantemir [2], Camelia Margareta Bogdănici [1,*], Daniel Constantin Brănişteanu [1,*] and Bogdan Doroftei [3]

[1] Department of Ophthalmology, "Grigore T. Popa" University of Medicine and Pharmacy, 700115 Iasi, Romania; anton.nicoleta1@umfiasi.ro (N.A.); ciprian.danielescu@umfiasi.ro (C.D.); anisia-iuliana.alexa@umfiasi.ro (A.I.A.)

[2] Oftaprof Ophthalmology Clinic, 700327 Iasi, Romania; dorin.chiselita@gmail.com (D.C.); alina.cantemir@gmail.com (A.C.)

[3] Mother and Child Medicine Department, Faculty of Medicine, "Grigore T. Popa" University of Medicine and Pharmacy, 16 Universitatii Street, 700115 Iasi, Romania; bogdandoroftei@gmail.com

* Correspondence: roxana-elena.ciuntu@umfiasi.ro (R.E.C.); camelia.bogdanici@umfiasi.ro (C.M.B.); daniel.branisteanu@umfiasi.ro (D.C.B.)

**Abstract:** Sleep apnea syndrome (SAS) is a condition characterized by recurrent episodes of total or partial collapse of the upper respiratory tract associated with daytime drowsiness that cannot be explained by other factors. SAS is a pathology that can cause ophthalmological damage both directly through the pathophysiological mechanism characteristic of the disease on the ocular system, and indirectly by promoting the development of other pathologies (cardiovascular, metabolic), which are a risk factor for ocular morbidity in the absence of sleep apnea syndrome. The aim of this paper is to highlight the ocular symptoms determined by sleep apnea syndrome (SAS), by analyzing literature over the past 20 years. Method: A mini-review that collected data from Pub Med Central, ResearchGate, GoogleScholar, DovePress, ScienceDirect, Elsevier, related to the ocular implications given by sleep apnea syndrome, with or without continuous positive airway pressure (CPAP) treatment. The study included articles that identified a number of eye conditions associated with sleep apnea, such as: dry eye syndrome and impaired ocular surface, glaucoma, non-arteritic anterior ischemic optic neuropathy, floppy eyelid syndrome, keratoconus, central serous chorioretinopathy, central vein occlusion, corneal neovascularization, and age-related macular degeneration. Sleep apnea syndrome is a pathology that can cause the onset or worsening of varying degrees of severity eye diseases by its pathophysiological mechanism, with a different impact on the quality of the individual's life. On one hand, the purpose of this review is to identify studies in literature that associate sleep apnea syndrome with eye alterations; on the other hand, to inform the Romanian medical staff in different fields of the patients' guidance diagnosed with SAS to an ophthalmology clinic since early and mild symptoms, so that these patients benefit from an ophthalmological approach and monitoring, in an attempt to diagnose and treat eye diseases in time and prevent their worsening.

**Keywords:** eye diseases; dry eye syndrome; obstructive sleep apnea; glaucoma; ischemic optic neuropathy; diabetic retinopathy; continuous positive airway pressure

## 1. Introduction

Sleep apnea syndrome (SAS) is a condition characterized by recurrent episodes of total or partial collapse of the upper respiratory tract (lasting at least 10 s and a frequency greater than 10 per hour), associated with daytime drowsiness that cannot be explained by other factors, or which "experience two or more of the following factors: episodes of suffocation during sleep, recurrent episodes of awakening during sleep, fatigue during the day, impaired ability to concentrate, restless sleep" [1,2]. The complexity of the pathology, as well as the importance of its diagnosis and treatment as early as possible is motivated by the numerous complications it can cause, being a pathology that can disturb the balance of

several devices and systems. The severity of SAS can be estimated using the Apnea Hypopnea Index (AHI), apnea being defined by the total obstruction of the CRS, and hypopnea by a partial one. Thus, AHI highlights the number of episodes of apnea/hypopnea, being able to establish the lack of pathology (AHI < 5/h) or its presence (AHI ≥ 5/h), the latter being able to classify SAS as mild (AHI: 5–15/h), moderate (AHI: 1–30/h), and severe (AHI > 30/h) [3]. According to the third edition of the International Classification Sleep Disorder (ICSD-3) defines OSA as an index of obstructive respiratory distress (RDI) caused by PSG ≥ 5 events/h associated with typical OSA symptoms (e.g., refreshed sleep, daytime drowsiness, fatigue, or insomnia, waking up with a feeling of panting or suffocation, hard snoring, or control apnea) or an obstructive CDI ≥ 15 events/h (even in the absence of symptoms [4,5]. SAS represents a pathology could at can cause ophthalmological damage both directly, through the pathophysiological mechanism characteristic of the disease on the ocular system, and indirectly by promoting the development of other pathologies (cardiovascular, metabolic, etc.), which are a risk factor for ocular morbidity in the absence of sleep apnea syndrome. Several articles in literature have highlighted the impact that SAS has on developing new ophthalmologic pathologies or by worsening the pre-existing diseases. There are many studies that demonstrate the association between ocular pathology associated with obstructive sleep apnea syndrome [6–8]. The most common ocular manifestations found in association with SAS are dry eye syndrome and ocular surface damage, glaucoma, non-arteritic anterior ischemic optic neuropathy, floppy eyelid syndrome, keratoconus, central serous chorioretinopathy, neural occlusion, corneal vein, central venous occlusion, and age-related macular degeneration. Although until now there are only assumptions on the mechanism of occurrence of these ophthalmic complications, alterations in blood pressure and hypoxemia characteristic of SAS patients have been proposed as etiology for glaucoma, optic disc edema, the degree of diabetic retinopathy, and retinal vein obstruction (highlighted by Glacet-Bernand et al., who found a much higher risk compared to the control group in patients diagnosed with SAS to develop obstruction) [9–11]. Giannicola Iannella and colleagues in a meta-analysis correlate obstructive sleep apnea (OSA) with nasal function, involving both mucociliary clearance and olfactory function. The severity of sleep apnea was also related to the degree of olfactory dysfunction, with lower TDI scores in patients with higher rates of apnea. The sum of the results from each of three different sub-tests results in a total score defined as TDI. According to the existing literature, the results of the TDI score indicate hyposmia when the total TDI score is <30.5, anosmia when it is <16.5, and no OD when the TDI score is >30.5. A linear correlation between Apnea-Hypopnea Index (AHI) increase and TDI decrease was detected [12]. An Annalisa Pace study was conducted to evaluate the relation between OSA (obstructive sleep apnea) and allergic rhinitis (AR). Non-allergic rhinitis with eosinophilia syndrome (NARES) is a condition with a symptomatology apparently similar to AR. Results showed that 60% of patients affected by NARES presented OSA. In conclusion, data showed that there was an increased risk of OSA in NARES patients respect to AR patients and healthy patients [13].

## 2. Materials and Methods

### 2.1. Database Searches

A mini-review that collected data from the following databases: PubMed/Medline, ResearchGate, GoogleScholar, DovePress, ScienceDirect, Elsevier, and Cochrane Database of Systematic Reviews (CDSR). Pub Med Central, related to the ocular implications given by sleep apnea syndrome, with or without continuous positive airway pressure treatment. The study included articles that identified a number of eye conditions associated with sleep apnea, such as: dry eye syndrome, glaucoma, non-arteritic anterior ischemic optic neuropathy, floppy eyelid syndrome, keratoconus, central serous chorioretinopathy, central vein occlusion, corneal neovascularization, and age-related macular degeneration

### 2.2. Eligibility Criteria

The study included articles that identified a number of eye conditions associated with sleep apnea, such as: dry eye syndrome, glaucoma, non-arteritic anterior ischemic optic neuropathy, floppy eyelid syndrome, keratoconus, central serous chorioretinopathy, central vein occlusion, corneal neovascularization, and age-related macular degeneration. A mini-review that collected data from Pub Med Central, related to the ocular implications given by sleep apnea syndrome, with or without continuous positive airway pressure treatment. Studies reported in English and in other languages, case reports, reviews studies not older than 2000 were selected. Conference posters, computational simulations, and letters to the editor were excluded.

### 3. Results

#### 3.1. Study Selection

We identified a total of 47 articles that identified a number of eye conditions associated with sleep apnea.

#### 3.2. Dry Eye Syndrome and Damage to the Eye Surface

A first complication of SAS is the dry eye syndrome, especially in those associated with floppy eye syndrome, highlighted by the Schirmer 1 test and the break-up time test. Vehof et al. conducted a cohort study in 2020, in which they confirmed the connection between the dry eye syndrome and other factors, including ODS. High blood pressure and high BMI were strongly associated with the less dry eye or current smoking, while former smokers had drier eyes [14]. The inflammation of the ocular surface specific to the dry eye syndrome develops as a result of increased tear osmolarity, proinflammatory cytokines accumulation secreted by the lacrimal gland over the ocular surface, and the delayed clearing by tears [15]. In SAS, the levels of proinflammatory cytokines (TNF-alpha, IL-1, IL-6) are increased based on chronic intermittent hypoxia. As AHI increases, the mechanical stress of tissues, the hypoxia and inflammation of the ocular surface also increase, which in turn lead to the loss of meibomian and goblet cells function, decreased corneal sensitivity, and reduced tear production in response to the stimulation of the tear glands. Karaca et al. (2019) showed that the damage to the meibomian glands in the upper eyelids in patients with SAS was much more pronounced compared to healthy subjects, especially those with severe SAS. This rubbing-like effect could also be a cause of meibomian gland distortion in patients with SAS [16]. Figure 1 shows the mechanisms involved in producing ocular changes in patients with SAS.

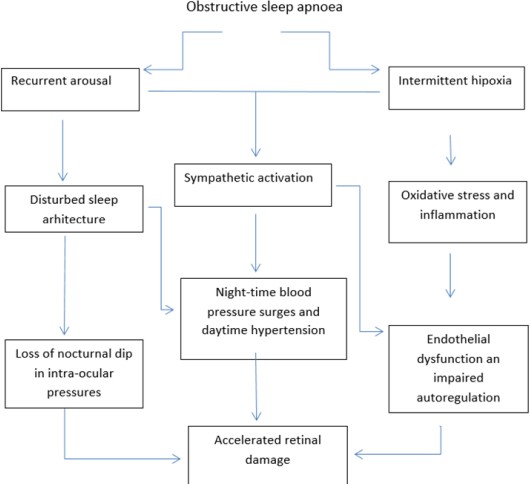

**Figure 1.** Mechanisms involved in SAS that cause eye damage.

### 3.3. Floppy Eyelid Syndrome

It is the syndrome in which the upper eyelids are easily reversed at upward traction, leading to events related to the exposure of the ocular surface. In literature, the emergence of FES in patients with SAS (incidence from 2 to 40%) varies considerably. The study conducted by Karaca et al. in 2019 established a prevalence of 22.2% in patients with severe SAS [16]. A prospective study with 127 patients found that 25.8% of the patients with SAS also had FES, and this percentage continues with up to 40% in patients diagnosed with severe SAS [7]. Regarding the mechanism of SAS, and not the histology in the eyelid, in addition to chronic inflammation present without atrophy in the tissue, it was noted that at that level, there is a transient ischemic attack, which is highlighted by the hypoxia present during the recurrent episodes of apnea specific to SAS.

### 3.4. Keratoconus

It is an ocular pathology with bilateral damage defined by the progressive thinning of the cornea, the patients manifesting the occurrence of a cone at the level of the central portion, given by its protrusion outward, associating a refractive error, i.e., the irregular astigmatism [10]. By highlighting hypoxia and reperfusion injury as a mechanism for the development of keratoconus, the hypothesis of the SAS role in determining this pathology was proposed. A study conducted in the US on 362 patients diagnosed with keratoconus over a period of 14 years was identified, of which 101 filled in the Berlin questionnaire for SAS, which resulted in a 18% prevalence of AS, 47% of the remaining patients manifested an increased risk of SAS [3]. In a prospective wide case-control with 616 patients diagnosed with keratoconus registered along with 616 control patients, having the same age, sex, BMI, on which the questionnaire Berlin was performed, there was a high record of SAS 29% versus 0.6% and a higher risk of SAS: 12.3 versus 6.5. In addition, it has been observed that patients with severe SAS had a more complicated type of keratoconus compared to other patients [3]. Studies evaluating the association between SAS and KCN have reported SAS as having a personal history of patients below 3 to 24% and that the risk of developing SAS varies between 7 and 53%. It is estimated that SAS is 5 times more common in patients with KCN than in the general adult population and that about half of patients with KCN have undiagnosed SAS, probably because the mild type of this pathology is the most common.

### 3.5. Corneal Neovascularization

The emergence of corneal neovascularization as an ophthalmic complication of SAS was introduced in 2017 by Konstantinous et al., who published the first case report of corneal neovascularization in a patient with SAS treated with continuous positive airway pressure (CPAP)—a device using a mask connected to an air pump, which prevents the collapse of respiratory airways. They reached the following conclusion: Patients wearing a CPAP mask not worn correctly are at risk of developing abnormal vessels in the cornea and exudate development in the lower part, a hypothesis explained by the fact that the incorrect positioning of the CPAP mask will promote the development of a pressure-induced surface trauma [17].

### 3.6. Age-Related Macular Degeneration (ARMD)

It is one of the leading causes of central vision impairment in patients over 50 years old in developed countries, being the leading cause of irreversible vision loss in elderly patients, and responsible for 8.7% of blindness in the world [18]. The fact that one of the factors underlying the pathophysiology of neovascular ARMD is vascular endothelial growth factor (VEGF), determines that its inhibition is the basic therapy, called anti-VEGF therapy, approved in 2006. The emergence of anti-VEGF therapy determined "the removal of neovascular ARMD from the list of confirmed pathologies" [19]. The connection between ODS and ARMD was confirmed by Keenan et al. in a study published in 2016 involving a cohort of patients diagnosed with SAS (67.786 cases) and a cohort of ARMD patients (248.408 cases) [20]. Due to the fact that the risk factors for developing ARMD include

elements that may occur in SAS, due to the systemic nature of the disease, factors such as cardiovascular (atherosclerosis, hypertension, heart failure, dyslipidemia), metabolic (diabetes, obesity), pharmacological (used in comorbidities associated with ODS: therapy with diuretics, antihypertensives, anti-inflammatory), the risk of developing ARMD in SAS may be suggested [21]. In 2016, Schaal et al. conducted a study in which they wanted to highlight the effect of treating SAS patients with CPAP on the evolution of ARMD and on the effectiveness of anti-VEGF treatment. Following their study, they highlighted the beneficial effects that SAS treatment has on the exudative ARMD therapy. The CPAP and anti-VEGF therapy combination had beneficial effects in approximately $40 \pm 17$ weeks after initiation of therapy. Schaal et al. concluded that the non-treatment of SAS might contribute to the lack of exudative ARMD response to anti-VEGF treatment [22].

### 3.7. Diabetic Retinopathy (DR)

It is the most common microvascular complication of diabetes and a major cause of vision loss. In general, the prevalence of DR, the proliferative type and diabetic macular edema in the general population is estimated at 34.6%, 7% and 6.8%, respectively [23]. SAS affects between 58% and 86% of patients diagnosed with diabetes. The common element between SAS and type 2 diabetes mellitus (DM2) is obesity. The connection between type 2 diabetes and SAS has been highlighted in several studies suggesting that the prevalence of SAS is doubled in patients with ophthalmic complications of type 2 diabetes, compared to those with type 2 diabetes without complications. One of the mechanisms proposed for this is the alteration of glycemic control, caused by inflammation and oxidative stress characteristic of SAS, the type of final glycosylation products, elements that are also found in the pathophysiology of vascular complications characteristic of diabetes [24]. The most recent study was conducted in 2020 by Adderley et al., according to which they concluded that patients with diabetes, especially with type 2 diabetes, have a higher risk of developing microvascular and cardiovascular complications than those with type 1 diabetes. There is a 50% higher frequency of developing comorbidities (congestive heart failure, myocardial infarction, stroke, ischemic transient accident, 18% of developed BCR, and other comorbidities), compared to the group control that consisted of patients with diabetes, but who were not diagnosed with SAS [25]. Zhang et al. also confirmed the influence of SAS on the complications of type 2 diabetes [26] regarding the effect of CPAP on DR, studies which have reached different conclusions: One study conducted in the UK has revealed a significant improvement in macular edema or fundus examination and another one that highlighted the fact that people who received CPAP treatment were less likely to develop pre-proliferative or proliferative DR [3].

### 3.8. Central Retinal Vein Occlusion (CRVO)

It is another condition that is a common cause of blindness after diabetic retinopathy. Patients generally have other associated pathologies—cardiovascular, coagulation disorders—and are treated with intravitreal injections with anti-vascular endothelial growth factor to prevent the occurrence of neovascularization in the retina. Although the first connection of CRVO with SAS was reported in 2009 by Leroux les Jardins et al. [27], more recently, in 2009 and 2014, two published cases of a rarer retinal vein occlusion with bilateral involvement of central vein have been described in patients with SAS, morbid obesity and high hematocrit [28,29]. In one of these studies, elevated levels of fibrinogen and C-reactive protein were highlighted, suggesting a state of chronic inflammation along with a regulatory disorder of thrombogenic mechanisms [30]. Another study on patients with CRVO revealed a prevalence of SAS in 77% of the patients selected based on nocturnal symptoms for screening [7]. Glacet–Bernard et al. have identified the strongest connection between the two diseases, by polysomnography highlighting a SAS 37% prevalence among patients with CRVO, significantly increased compared to the control group (11%). Following polysomnography, it was observed that AHI and oxygen saturation levels are significantly higher in patients with CRVO [27,31]. The explanation of the two pathologies

connection is given by the intermittent nocturnal hypoxia characteristic of SAS which increases oxidative systemic stress and the production of oxygen free radicals and inflammatory cytokines such as IL-1 and IL-6 which activates the extrinsic coagulation pathway and subsequently venous thrombosis in SAS patients, which acts as a trigger for CRVO development. SAS can subsequently act as a trigger in those with CRVO for the development of other comorbidities such as atherosclerosis, DM, and hypertension [27,29,32].

### 3.9. Central Serous Chorioretinopathy (CSCR)

The disease is characterized pathogenically by a serous detachment of the retina, most often centrally located, especially in the macula, caused by the accumulation of serous fluid under the retina, and clinically by the emergence of abnormalities in the visual field, darkness, with or without magnifying images, visual distortions (patients perceive straight lines as wavy) [7,30]. In some studies, the connection between CSCR and SAS is suggested based on increased plasma levels of catecholamines. In patients with SAS, there is a high level of norepinephrine and plasma adrenaline, and this increase in sympathetic tone causes an increase in endothelial dysfunction at the retinal-blood barrier, which leads to the accumulation of serous fluid at the subretinal level [7,14]. In a recent meta-analysis, Huon et al. noticed a link between SAS and CSCR, following the analysis of two case-control studies [9]. A smaller study with 23 patients diagnosed with CSCR showed that of these, 14 patients had SAS (60.9%) [33]. This hypothesis is also supported by a number of presented cases: Jain et al. presented the case of a 45-year-old male patient, whose clinical condition characteristic of CSCR improved after CPAP treatment, on the principle of decreasing sympathetic tone [34]. Another meta-analysis conducted by Wu et al. in 2018 highlighted that there is no significant difference in terms of choroidal thickness between mild SAS and control patients, but that patients with moderate and severe symptoms have a thinner choroidal compared to the control group [35].

### 3.10. Non-Arteritic Anterior Ischemic Optic Neuropathy (NAION)

It is a severe ischemic ocular impairment, involving the anterior segment of the optic nerve and clinically associated with pupillary deficit and papillary edema. NAION represents a pathology with a vascular pathophysiological mechanism, similar to SAS, but which could not be fully explained. The pathology involves the emergence of ischemia in the posterior ciliary artery, which nourishes the optic nerve. The ischemia causes edema at the exit of the nerve from the eyeball with the emergence of compression at that level, causing vascular compromise. Patients with SAS have impaired self-regulation of blood circulation and imbalance between nitric oxide and endothelin, which amplifies tissue hypoxia and hypoperfusion. In addition, in SAS, the intracranial pressure increases causing the limitation of the perfusion at the level of the optic nerve which can lead to NAION [7,36,37]. In 2015, Wu et al. performed a meta-analysis that included studies published between 2002 and 2013, which followed the connection between ODS and NAION. The study confirmed the link between the two pathologies, compared to control patients, SAS patients having a 6 times higher frequency for NAION [36]. Stein et al. highlighted that patients with SAS have a 16% higher risk of developing NAION compared to patients who have not been diagnosed with this sleep pathology [7,30,38]. Regarding the effect of SAS on NAION CPAP therapy, a 3-year study in NOIAN patients showed that those who associate severe non-compliant SAS with CPAP treatment "have a 5.5 times greater risk of damage to the contralateral eye, compared to patients who do not have SAS or who have moderate SAS without any indication for CPAP". This study suggested that CPAP treatment of SAS might be considered a therapeutic strategy in severe SAS to reduce the risk of bilateralization of NAION [36].

### 3.11. Glaucoma

Glaucoma is a progressive optic neuropathy that associates accelerated apoptosis of the lymph node cells in the retina. The classification of different types of glaucoma depends

on the anatomical condition of the iridocorneal angle and the etiology of glaucoma (primary or secondary). There are several types of glaucoma, but those related to SAS are normal blood pressure glaucoma and primary open-angle glaucoma [7,30,39]. The prevalence of glaucoma in SAS patients has been investigated by numerous studies, the results ranging from 5.9% [40], 12.9% (Muniesa et al., 2013) [41], 27% (Bendel et al., 2008) to 30% (Nesreen et al., 2019 [42]. The connection between SAS and glaucoma has been confirmed over time by several studies, which investigated several glaucoma-specific ocular parameters, including: increased IOP, thinning of SFNR (retinal nerve fiber layer) peripapillary, field impairment visually, and glaucomatous changes of the optic nerve [43].

SP Hashim et al. state that patients who do not have glaucoma but have SAS may develop glaucoma if SAS is not well controlled. Thus, strict control of this condition can contribute to a better management of glaucoma and stabilization of visual fields [44]. In 2019, Abdullayev et al. conducted a study for the evaluation of SFNR and lymph node complex in patients with SAS, which confirmed that 40% of retinal lymph node axons are already lost before visual field defects [45], Shinmei et al. tried to see the influence of apnea on the IOP; therefore, they examined the immediate dynamic response of the parameter in relation to the altered respiratory events in patients' sleep. "The mean IOP was lower in patients free of apnea". They concluded that "the variation in IOP in patients with SAS is influenced to some extent by the circadian rhythm, while apnea and hypopnea occur accidentally during sleep, regardless of the evolution of IOP". Only obstructive sleep apnea influences the value of intraocular pressure, in terms of keeping breathing [46]. Lee et al. published an article investigating the association between SAS and glaucoma in young adults. They evaluated a sample of 858 patients aged 19–21, all healthy, who were re-evaluated at ages 21–24, 20% of whom were diagnosed with SAS. Following this study, they concluded that patients diagnosed with SAS had an increase in SFNR of approximately 4 μm (especially in the infratemporal and supratemporal segments), compared to the control group [43].

The change in the thickness of SNFR (retinal nerve fiber layer) was also highlighted by Huseyinoglu et al. (2014) and Lin et al. (2011) [47] in patients who had a severe type of SAS compared to the control group. In their studies, the authors consisted at a thinning in the early SNFR superolateral time, these two sectors being most severely affected in SAS. Thinning of SFNR is a change found in early glaucoma, which may indicate an increased risk of SAS- associated glaucoma from a young age [43].

In our recent prospective study of 65 eyes in 65 patients, the association of glaucoma with the severity of sleep apnea syndrome was demonstrated. Within the studied group, patients with mild or moderate primary open-angle glaucoma, with moderate or severe dry-eye syndrome, patients with floppy-eyelid syndrome, with optical non-arteritis ischemic neuropathy, and a patient with retinal central vein occlusion were identified. The increased rate of the apnea syndrome during sleep produces a severe disorder of the ocular surface and a retinal neuro- degenerative disorder [8].

Regarding the role of SAS treatment by CPAP in the variation of intraocular pressure (IOP), there are several contradictory studies: Some claim that this treatment causes increased IOP [5], while others claim that it would help prevent glaucoma and optic nerve neuropathy [37]. The results showed that during the apnea and hypopneic episodes the IOP decreased, the decrease being influenced by the negative intrathoracic pressure, determined by the inspiratory test, despite the fact that there is a blockage at the CRS level [22,39]. Mentek et al. point out that the CPAP treatment restored the circadian rhythm of IOP in 67% of apnea patients who had an abnormal rhythm, pointing out the role played by sleep cycles on IOP variation. It is suggested that there is a direct effect of nocturnal inspiratory efforts on the decrease in IOP observed in SAS patients [30].

## 4. Studies Limitations

The small number of references associating sleep apnea and ophthalmological pathology, despite the long follow-up period.

## 5. Conclusions

SAS is a pathology that can cause the onset or worsening of eye diseases with different degrees of severity through its pathophysiological mechanism, with a different impact on the individual's quality of life. The connection between SAS and glaucoma has been confirmed over time by several studies: Patients who do not have glaucoma, but have SAS, may develop glaucoma if SAS is not well controlled. In addition, a frequent connection confirmed between the two pathologies in numerous studies is represented by NAION, compared to control patients, SAS patients presenting a frequency 6 times higher for NAION.

A large part of the published articles on sleep apnea, especially in Romania, focused less on the relationship between SAS and ophthalmic diseases and more on the relationship between SAS and other pathologies: cardiovascular (hypertension), metabolic (obesity, diabetes), and cerebrovascular. The role of this review was to inform the medical staff in different fields of the patients' guidance diagnosed with SAS to an ophthalmology clinic since early and mild symptoms, so that these patients will benefit from one ophthalmological approach and monitoring, in an attempt to diagnose and treat eye diseases in time and prevent their worsening.

**Author Contributions:** Conceptualization, N.A. and R.E.C.; methodology, N.A., B.D., C.M.B., R.E.C., A.I.A., A.C., C.D., D.C. and D.C.B.; validation, N.A., R.E.C., D.C. and B.D.; investigation, N.A., R.E.C., B.D., D.C.B. and C.M.B. All authors have read and agreed to the published version of the manuscript.

**Funding:** The work did not use financial resources involving any company or institution.

**Institutional Review Board Statement:** Not applicable.

**Informed Consent Statement:** Not applicable.

**Data Availability Statement:** All data generated or analyzed during this study are included in this published article.

**Acknowledgments:** Professional editing, linguistic, and technical assistance performed by Irina Radu, Individual Service Provider, certified translator in Medicine and Pharmacy (credentials: E0048/2014).

**Conflicts of Interest:** The authors declare that they have no conflict of interest.

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
