# Peer review of "Ocular Implications in Patients with Sleep Apnea"

_applsci, doi:10.3390/app112110086_

Round 1
Reviewer 1 Report
The role of this review is to inform the medical staff in different fields of the patients’ guidance diagnosed with SAS to an ophthalmology clinic since early and mild symptoms, so that these patients benefit from anophthalmological approach and monitoring, in an attempt to diagnose and treat eye diseases in time and prevent their worsening.
A very interesting concept and study.
Well done!
Reviewer 2 Report
Introduction
Line 46, please better define OSAS criteria according to AASM guidelines and cite doi:10.5664/jcsm.6506.
line 49, More and more evidence correlates obstructive sleep apnea (OSA) with nasal function both involving mucociliary clearance and olfactory function. In addition, nasosinus diseases such as allergic and vasomotor rhinitis could be considered a risk factor for sleep apnea. The severity of sleep apnea was also related to the degree of olfactory dysfunction, with lower TDI scores in patients with higher apnea rates and please cite doi:10.1007/s00405-020-06316-w. and doi:10.3390/medicina56090454 line 55 prefer could at can in all the sentences when you express and hypothesis line 66 please report always references when define a concept as ''Cristescu TR, Mihălțan FD. Ocular pathology associated with obstructive sleep apnea syndrome. Rom J Ophthalmol. 2020;64(3):261-268. Methods please report a flow diagram on literature review protocol section 2.3 could be reported in results section and cite all the articles added in the paper Results line 102, add here the reference edit ''et al.'' in italics in all the text please add a flow chart on ocular disease pathophysiology and osas Please add a section ''Studies Limitations''Author Response
Please see the attachment!
